# Assessment of Anxiety, Depression, Work-Related Stress, and Burnout in Health Care Workers (HCWs) Affected by COVID-19: Results of a Case–Control Study in Italy

**DOI:** 10.3390/jcm11154434

**Published:** 2022-07-29

**Authors:** Giuseppe La Torre, Vanessa India Barletta, Mattia Marte, Francesca Paludetti, Augusto Faticoni, Lavinia Camilla Barone, Ilaria Rocchi, Filippo Picchioni, Carlo Maria Previte, Pasquale Serruto, Gloria Deriu, Camilla Ajassa, Roberta Campagna, Guido Antonelli, Claudio Maria Matroianni

**Affiliations:** 1Department of Public Health and Infectious Diseases, Sapienza University of Rome, 00185 Rome, Italy; vanessaindia.barletta@uniroma1.it (V.I.B.); mattia.marte@uniroma1.it (M.M.); francesca_paludetti@libero.it (F.P.); augusto.faticoni@uniroma1.it (A.F.); laviniacamilla.barone@uniroma1.it (L.C.B.); ilariarocchi16@gmail.com (I.R.); picchioni.1697595@studenti.uniroma1.it (F.P.); carlomaria.previte@uniroma1.it (C.M.P.); pasqualese@live.it (P.S.); gloria.deriu1@gmail.com (G.D.); camilla.ajassa@uniroma1.it (C.A.); claudiomaria.mastroianni@uniroma1.it (C.M.M.); 2Department of Molecular Medicine, Policlinico Umberto I, Sapienza University of Rome, 00161 Rome, Italy; roberta.campagna@uniroma1.it (R.C.); guido.antonelli@uniroma1.it (G.A.)

**Keywords:** anxiety, depression, work-related stress, burnout, health care workers, COVID-19, case–control

## Abstract

This study aims to investigate whether HCWs infected with COVID-19 may experience potential psychological consequences and a higher incidence of depression, anxiety, work-related stress, and burnout compared to non-infected HCWs. A case–control study with 774 participants was conducted comparing COVID-19-infected HCWs (cases) and non-infected HCWs (controls) from the Occupational Medicine Unit at the Teaching Hospital Policlinico Umberto I, who were administered the same questionnaire including Hospital Anxiety and Depression Scale, Copenhagen Burnout Inventory and Karasek’s Job Content Questionnaire. No differences in the levels of burnout and decision latitude were found between the two groups. Cases showed higher level of anxiety and job demand compared to controls. In contrast, levels of depression in the case group were significantly lower compared to the control group. The results are indicating the need for workplace health promotion activities based on stress and burnout management and prevention. Multiple organizational and work-related interventions can lower the impact of mental health-related issues in the COVID-19 pandemics, including the improvement of workplace infrastructures, as well as the adoption of correct and shared anti-contagion measures, which must include regular personal protective equipment supply, and the adoption of training programs that deal with mental health-related issues.

## 1. Introduction

COVID-19 pandemic has had a tremendous impact on all social and health care sectors, leading to unprecedented challenges for humanity. In this emergency, a great deal of attention has been focused on the medical complications caused by COVID-19. Recently, the potential direct effects on mental health in the population have received some attention, even if knowledge about this topic is limited [1,2,3].

Since the outbreak of the pandemic, healthcare professionals have played a crucial role, facing overwhelming and unforeseen changes in their regular duties. Within the space of a few days or few weeks, medical workers found themselves coping with the implementation of new organizational measures, resource management, and ongoing changes [4,5]. 

There is sufficient evidence that mental health problems have affected healthcare workers’ health during this pandemic [6,7,8]. 

Many studies highlighted how frontline HCWs experienced increased levels of mental health problems: fear of transmitting the infection to family members, higher perceived stress load, poor sleeping quality, burnout, depression, anxiety, post-traumatic stress disorder (PTSD) symptoms, psychological distress, and somatization [9,10,11]. Luo et al. (2020) carried out a systematic review, including 9207 studies conducted in 17 countries, which evaluated the psychological and mental impact of coronavirus disease on medical staff and the general population. According to the results of this study, a similar incidence of anxiety and depression was reported in both groups. In contrast, several studies carried out in China, Italy, Turkey, Spain, and Iran highlighted a higher incidence of psychological distress among HCWs [12]. As reported by scientific literature, the onset of depression, stress, and PTSD symptoms should not be underestimated, especially when involving medical workers. These symptoms can have significant consequences, leading to therapeutic errors and lower standards of treatment [13]. Therefore, paying special attention to the potential onset of psychiatric symptoms among HCWs is extremely important. To address this issue, the implementation of psychological training to prevent the effects of exposure to traumatic events appears to be fundamental.

The introduction of social support and educational interventions related to coping and resilience training can positively affect and strengthen the mental health wellbeing of health care personnel [14,15]. In this context, identifying possible risk factors associated with psychological problems among medical professionals is essential. Many studies showed that the following risk factors were associated with psychological distress: female gender, nursing professionals, victims of a stigma, exposure to infected patients, quarantine isolation [16], young age, limited work experience, work overload, unsafe workplace, lack of training and social support [17], work-related stress, and financial concerns [15]. 

According to the job demands-resources (JD-R) model [18], job characteristics can be classified into two broad categories, such as job demands and job resources. As far as concerns the first issue, these are the aspects of the job that require sustained effort, and are associated with certain costs. On the other hand, job resources are those aspects that are functional in dealing with job demands and are associated with personal growth and development. These job categories are associated with two different psychological processes. In other words, high job demands, associated with sustained effort, may lead to the exhaustion of employees’ resources and energy depletion, as well as health problems [19].

In contrast, the following factors were associated with a lower incidence of psychological symptoms: personal and organizational support, perception of control, positive attitude in the workplace, being sufficiently informed about the pandemic, protective equipment, training, and adequate resources [11], protective measures in the workplace [20]. Furthermore, it is fundamental to evaluate if having been infected with COVID-19 may lead to higher levels of depression, stress, and anxiety.

In this regard, two studies demonstrated that 96.2% of COVID-19 patients experienced a higher incidence of PTSD symptoms and significantly higher levels of depressive symptoms [21]. There is evidence that mental issues related to the COVID-19 pandemics, such as anxiety, depression, PTSD, and sleep disorders, have a great impact on healthcare workers, with a particular involvement of those who work on the frontline [22]. 

Hyeon-Ah Lee et al. (2021) demonstrated that healthcare workers quarantined for being a contact with COVID-19 patients had a lower level of depression in comparison to HCWs working in cohort wards and who were socially discriminated against. These authors found a path analysis model clearly showing that the predictors of depression in healthcare workers during the early stage of the COVID-19 pandemic are work-related stress and generalized anxiety. Moreover, burnout, considered a condition characterized by feelings of energy depletion or exhaustion, increased mental distance from one’s job, and reduced professional efficacy can be included in the path between stress and depression [23].

Seong and colleagues found that during the 2015 outbreak of the Middle East Respiratory Syndrome in Korea, healthcare workers that came in contact with patients with MERS and were quarantined at home had a lower rate of symptoms such as depression (*p* < 0.001) and acute stress disorder than the group who were in-hospital quarantined [24].

Tan et al. (2020) carried out a study to assess the immediate psychological impact on the workforce who returned to work after lockdown and quarantine in China. The participants reported a low prevalence of anxiety, depression, and stress [25].

The aim of this study is to investigate if HCWs who were infected with COVID-19 may experience, as a result of isolation, social rejection, physical changes produced by the virus, potential psychological consequences, and a higher incidence of depression, anxiety, work-related stress, and burnout compared to HCWs who were not infected with COVID-19. From the occupational point of view, this issue is relevant in terms of workplace health promotion as well as for increasing specific training for tackling mental health problems.

## 2. Materials and Methods

### 2.1. Study Design, Participants, and Procedures

This research was undertaken through a prospective case–control study, carried out between April 2020 and March 2021 at the Teaching Hospital Policlinico Umberto I in Rome, a big hospital with 1200 beds and 4500 HCWs. Cases included HCWs who were infected with SARS-CoV-2, whereas the controls were HCWs who were not infected. Case group participants were enrolled during a follow-up visit at the Occupational Medicine Unit of the Teaching Hospital after recovering from COVID-19, whereas the control group participants were enrolled during regular check-ups.

The inclusion criteria were as follows: (a) Cases and controls were individuals without cognitive impairment or any psychiatric illnesses previously diagnosed; (b) they were HCWs or Residents from all the Departments of the Teaching Hospital. Individuals were excluded if they were cooperative members, or workers of outsourcing companies (such as workers of the company canteen, fire service, or surveillance personnel). Cases and controls were matched by age and gender.

### 2.2. Measures

Both cases and controls were administered the same questionnaire with the aim to evaluate levels of depression, anxiety, work-related stress, and burnout. Patients were asked to self-fill in the questionnaires. Then, after the researchers checked the completeness of the answers, data were entered into a database in Excel.

The questionnaire included three tests: Hospital Anxiety and Depression Scale (HADS), Copenhagen Burnout Inventory (CBI), and Karasek’s Job Content Questionnaire (JQC). Moreover, the following socio-demographic data were collected: age, gender, marital status, having children, state of good health, professional role. Finally, the last question was “Do you rate your health as good?” (possible answers: Yes and No).

The HADS was used to measure anxiety and depressive symptoms. This self-report questionnaire is a fourteen-item scale of which seven of the items relate to anxiety and seven relate to depression. Participants were suggested to choose the option that best represented their emotional state, to answer as quickly as possible, and to refrain from changing their answers. This questionnaire enabled the creation of a self-report symptoms inventory, matching the psychological and psychiatric symptoms of each participant [26].

The CBI was used to evaluate personal, work, and patient-related burnout. The CBI consists of 19 items specifically evaluating personal-related (6 items), work-related (7 items), and client-related (6 items) burnout. An average score was calculated for each scale [27].

The JCQ was used to evaluate occupational stress. This questionnaire is based on a model theorized by Robert A. Karasek that postulates that the relationship between high job demand and low decision latitude leads to a state of perceived job strain. The job demand-control model postulated that the variables of demand and control are independent. Karasek’s questionnaire can be found in many different versions. In this study, a version including 15 questions was administered to the participants [28].

### 2.3. Statistical Analysis

Sample size calculations, using EpiCalc 2000, were based on the following assumptions:

Mean anxiety score in the control group: 10.00 (median score); mean anxiety score in the case group: 11.00 (increase of 10% with respect to control group); SD: 3.00; significance level: 0.05; power: 80%.

Using these assumptions, we calculated to recruit at least 141 HCWs in each group (282 overall).

Statistical analysis was performed using inferential and descriptive statistics. The descriptive statistical analysis of the sample was carried out respectively calculating: the average and the SD for quantitative variables and frequency, and percentages for qualitative variables. To estimate the reliability of the constructs of the questionnaires used, the Cronbach’s alpha was used. A value over 0.70 was considered satisfactory.

The Chi-square test was used to perform a univariate analysis to evaluate inter-group differences for qualitative variables. The Kolmogorov–Smirnov test was used to examine if variables were normally distributed. The Mann–Whitney U test or the Student’s *t* were used to compare the means of the two case and control groups of non-normally and normally distributed quantitative variables, respectively. A bivariate analysis was carried out to evaluate the presence of a correlation among variables. Since the variable distribution was found to be not normal, the Spearman correlation coefficient *r* was used. Except for age, all the bivariate analysis variables were included: anxiety levels, personal burnout, work burnout, client burnout, job demand, and decision latitude. Subsequently, a multiple linear regression, simultaneously analyzing more than one variable and type, was performed. In these models, the single quantitative variable was considered dependent, whereas all the others as covariate. The goodness of fit of the models was assessed using the R^2^. All statistical tests were performed using SPSS for Windows 25.0 (IBM, Armonk, NY, USA). A *p*-value < 0.05 was considered statistically significant.

## 3. Results

### 3.1. Sample Descriptive Data

A total of 774 participants were enrolled in our study and complied with the questionnaire administered (Table 1).

Among cases, 51 (13.3%) were enrolled in the period March–September 2020, and 333 (86.7%) in the period October 2020–March 2021. The response rate was 100% for both groups since the questionnaires were administered during the surveillance visit for all staff in the hospital, and filling them was mandatory.

The Cronbach’s alpha values were 0.927 for CBI (19 items), 0.828 and 0.743 for the anxiety and depression components of HADS (7 + 7 items), and 0.766 for JCQ (15 items).

Among all the subjects, 273 were men (35.3%), whereas 501 (64.7%) were women. The average age was 43.7 years (12.5 SD). Regarding marital status, 325 (42%) participants were either divorced, separated, single, or widower, whereas 449 participants (58%) were married or living with a partner. Of all the subjects, 420 (54.3%) had children whereas 354 (45.7%) did not. Regarding the state of health, 691 (89.3%) were in good health and 51 (6.6%) reported a poor state of health. Finally, 292 (37.7%) were doctors, whereas 295 (38.1%) and 159 (20.6%) were nurses and other HCWs, respectively.

### 3.2. Univariate Analysis

The univariate analysis (Table 2) highlighted that there were no significant differences in age, gender, marital status, having or not having children, state of health, or professional role. However, the case group (median 2, range 0–18) showed a significantly lower level of depression (*p* < 0.001) compared to the control group (median 3, range 0–20). The job demand variable showed a difference between groups (*p* = 0.060), with a higher workload found in the case group compared to the control. On the other hand, the anxiety variable also showed a difference between groups (*p* = 0.055) with higher levels among controls (median 6, range 0–20) compared to cases (median 5, range 0–19).

### 3.3. Bivariate Analysis

The Spearman correlation analysis reported the presence of a very strong positive correlation between anxiety and depression (r = 0.666; *p* < 0.001) implying a linear dependency between the two variables. Anxiety was also positively highly correlated to personal burnout (r = 0.675; *p* < 0.001), work burnout (r = 0.569; *p* < 0.001), and client burnout (r = 0.321; *p* < 0.001). A correlation was also found with job demand (r = 0.074; *p* = 0.043), and age (r = 0.114; *p =* 0.002) (Table 3). In addition, a positive correlation between depression and job demand (r = 0.083; *p* = 0.022) was observed. Age was found to be positively correlated with depression (r = 0.195; *p* < 0.001), implying that depressive symptoms worsen linearly as age increases. In contrast, age was observed to be negatively correlated with job demand and decision latitude (respectively r = −0.101 with *p* = 0.006 and r = −0.0324 with *p* < 0.001), meaning that workload and decision-making capabilities are increased among early career health professionals. A significant and positive correlation was found between age and personal burnout (r = 0.147; *p* = 0.048), work burnout (0.131; *p* < 0.001), and patient burnout (r = 0.151; *p* < 0.001), inferring that as age increases, HCWs experience higher levels of burnout. In addition, all types of burnout (personal, work, and patient) were found to positively correlate with depression (personal burnout r = 0.633; *p* < 0.001; work burnout r = 0.583; *p* < 0.001; client burnout r = 0.365; *p* < 0.001). Finally, burnout was observed to be significantly and positively associated with both job demand and decision latitude.

### 3.4. Multivariate Analysis 

A multivariate analysis (Table 4) was performed for the following variables: depression, anxiety, personal burnout, work burnout, client burnout, job demand, and decision latitude. A positive correlation was found between depression and personal burnout (B standardized coefficient = 0.155); work burnout (B = 0.127) client burnout (B = 0.093), age (B = 0.073), and anxiety (B = 0.453). The analysis highlighted a negative correlation between depression and the case group (B = −0.053; *p* = 0.043). Therefore, HCWs who were infected with COVID-19 showed significantly lower levels of depression compared to the control group. On the other hand, cases showed a high level of anxiety when compared to controls (B = 0.430; *p* < 0.001). In addition, a positive correlation between anxiety, personal burnout (B = 0.417) and depression was found (B = 0.430). Data from the present study showed a negative correlation between anxiety and decision latitude, implying that anxiety levels tend to decrease among medical professionals with long experience (B = −0.070) and higher decision latitude (B = −0.049). The multivariate analysis for personal and work burnout highlighted that these two variables were not positively correlated with the case group. In contrast, a negative association was found between patient burnout and the case group (B = −0.053; *p* = 0.09). Furthermore, a positive correlation was observed between the case group (B = 0.083; *p* = 0.014) and job demand, meaning that HCWs who were infected with COVID-19 experienced increased workload and work-related stress. Finally, no significant correlation between decision latitude and the variables cases/controls was observed. We tried also to perform analysis by gender, but this variable did not show effect modification (data not shown).

## 4. Discussion

According to the results of the present study, anxiety, depression, and burnout (work, personal, and patient) are significantly and positively correlated. In addition, work-related stress and burnout are linearly dependent variables. Our findings highlighted that the prevalence of depression and burnout increases proportionally with age. In contrast, younger HCWs experience higher levels of work-related stress. The statistical analysis reported no significant difference in decision latitude and personal and work burnout between cases and controls. The results of this study showed lower levels of depression in HCWs who were infected with COVID-19 compared to those who were not. However, a higher level of anxiety and job demand, which is an indicator of work-related stress, was found in the case group compared to the control group. Moreover, the case group experienced a significantly higher level of patient burnout compared to the control group. Our findings reveal that HCWs who were infected with COVID-19 experienced a lower level of depression compared to HCWs who were not infected. Vindegaard et al. (2020) [21] performed a systematic review whose results appeared to be in contrast with our results. In this study, it is reported that COVID-19 patients had a significantly higher level of depressive symptoms (29.2%) compared to those who had not been infected (9.8%) (*p* = 0.016). However, in this study participants came from the general population, and are not HCWs directly involved in the healthcare facilities. Moreover, the timing of the interview could have an influence. As specified in Section 2, in our study participants belonging to the case group were enrolled during a follow-up visit after recovering from COVID-19, and this could have led to the mitigation of the level of depressive symptoms due to the negativization of the nasal swab and the recovery.

We decided not to take the interview for cases during their positivity for several reasons, among which are the comparability of the administration of the questionnaire, and above all the intention to use a disability management approach for the following cases. The choice of measuring stress, depression, anxiety, and burnout after recovering from COVID-19 was strictly related to an occupational point of view. Our main interest was to assess whether or not HCWs who recovered were ready to start working again from a psychological perspective.

Considering the literature, no previous study has investigated the potential psychological consequences in HCWs who were infected with COVID-19. Scientific literature highlights how the psychological well-being of medical professionals has been dramatically affected since the outbreak of the COVID-19 pandemic. Health care personnel have experienced increased distress because of the fear of being infected and transmitting the infection to family members [9,10,11]. Concerning the possible difference between different type of HCWs, one can see that doctors have lower levels of depression compared to other healthcare professionals, doctors and nurses have the same level of personal and work burnout, while nurses show higher levels of job demand and lower level of decision latitude compared to other healthcare professionals. It is possible to hypothesize that lower levels of depression among medical professionals who contracted the infection and subsequently recovered could be linked to a reduced fear for their own and family health [29]. A recent systematic review carried out by Hannemann and coll. underlines there are several pandemic risk factors for mental health problems in HCWs, including as high risk and fear of infection. However, other factors such as resilience, active, and emotion-focused coping strategies, and social support can be considered beneficial when protecting different aspects of mental health in HCWs [30].

Consequently, HCWs who have been infected with COVID-19 are more likely to consider this event as a protective factor. In this context, contact tracing and surveillance have been operating at the teaching hospital Policlinico Umberto I since the early phases of the pandemic. This network operates by remotely monitoring HCWs who have tested positive, providing protocol guidelines and personal support. Scientific literature shows that organizational and personal support measures in the workplace have been associated with lower psychological stress [31,32]. It could be deduced that this network has favored psychological well-being among medical professionals who were infected with COVID-19.

### 4.1. Practical Applications

From a practical point of view, this study suggests the important role of the occupational physician in order to take under control the mental health of healthcare workers. The study revealed the need not only of assessing continuously the levels of stress, anxiety, depression, and burnout in healthcare workers, but also of addressing this issue with practical and educational intervention. From an occupational point of view, the results indicate the need for workplace health promotion activities based on stress and burnout management and prevention. There is sufficient evidence that multiple organizational and work-related interventions can lower the impact of mental health-related issues in COVID-19 pandemics. Among these interventions we must recall the improvement of workplace infrastructures, as well as the adoption of correct and shared anti-contagion measures, which must include regular personal protective equipment supply, and the adoption of continuous medical education programs that deal with mental health-related issues [22], and organization of online support services [33]. A very recent systematic review demonstrates that interventions based on evidence-based protocols, such as individual and group-based cognitive behavioral therapy (CBT) for post-traumatic stress disorders, anxiety, and depression, are capable of leading to reliable changes in mental health symptoms. [34].

Mental support of HCWs in this pandemic is fundamental, including the use of telemental health services [35]. A recent review by Schwartz and colleagues reported available tools that can be applied in time of COVID-19 pandemic, including the physician support line, a free and confidential support line service made up of volunteer psychiatrists, joined together to provide peer support for their physician colleagues; the National Suicide Prevention Hotline and Crisis Text Line which operate 24/7; Headspace, a popular mindfulness web-based application, is offering free membership to U.S.-based HCWs to help cope with stress and anxiety with resources for sleep, meditation, and movement exercises; the Accreditation Council for Graduate Medical Education (ACGME) AWARE Well-Being gives resources that include video workshops, podcasts, and a web-based application specifically designed for promoting wellbeing in the graduate medical education community; the Well-Being in the Time of COVID-19 podcast by Stuart Slavin, a ACGME’s Senior Scholar for Wellbeing, which provides well-being strategies for residents, fellows, and other clinicians.

This review underlines also the importance of available literature resources to support HCWs such as those provided by the websites of the National Academy of Medicine, the Centers for Disease Control and Prevention, the Center for the Study of Traumatic Stress, and the United Kingdom’s Intensive Care Society [36]. Gray et al. developed the concept of “Mental Health PPE”, indicating that personal protective equipment (PPE) for employees’ mental and emotional health is just as important as physical protective equipment. They developed a program based on two different teams: (1) mental health liaisons for delivering preventive support to COVID-19 hospital units and emergency departments; and (2) mental health crisis response teams, available 24 h a day for supporting and mitigating staff crises as needed [37]. Finally, Oldham et al. demonstrated that team-based proactive consultation-liaison psychiatry is capable of enhancing HCW satisfaction and reducing burnout [38].

### 4.2. Strengths and Limitations

This study presents two main strengths: a large sample number of 774 participants and the collection of information about anxiety, depression, burnout, and work-related stress in real time avoided any type of recall bias.

However, this study has some limitations: all the participants work in the teaching hospital Policlinico Umberto I. Therefore, it could be difficult to apply the results of the current study to the general population. A second limitation is represented by the modalities of the questionnaire administration. The tests were administered in the presence of a doctor, a condition that may have influenced the HCWs response. However, since both cases and control group participants were administered the test following the same modalities (health surveillance visit), a risk of misclassification can be excluded. Levels of depression, anxiety, work-related stress, and burnout were found to be significantly higher among HCWs who participated in this research. Another possible limitation of this study was concerning variables that could have an impact on the results, such as smoking status, alcohol drinking, and substance abuse, which were not included since a lot of missing data was present. Finally, we did not take into account the severity of symptoms when cases were infected.

## 5. Conclusions

This study found in HCWs recently infected with SARS-CoV-2 low levels of depressive symptoms, and high levels of anxiety and job demand if compared to HCWs who were not infected. These findings could represent an opportunity to further investigate this phenomenon often underestimated. Future research on this issue is needed, given the probability of high mutation rates of the RNA viruses such as SARS-CoV-2 [39].

## Figures and Tables

**Table 1 jcm-11-04434-t001:** Sample descriptive data.

Variable	Number (%) or Mean (Standard Deviation)
Gender	
*Female*	501 (64.7)
*Male*	273 (35.3)
Age	43.71 (12.557)
Marital status	
*Divorced*/*Separated*/*Single*/*Widower*	325 (42.0)
*Married*/*Living with a partner*	449 (58.0)
Children	
*Yes*	420 (54.3)
*No*	354 (45.7)
State of good health	
*Yes*	691 (89.3)
*No*	51 (6.6)
*Missing*	32 (4.1)
Professional role	
*Doctor*	292 (37.7)
*Nurse*	283 (36.5)
*Midwifery*	12 (1.6)
*Ancillary worker*	131 (16.9)
*Technicians*	28 (3.6)

**Table 2 jcm-11-04434-t002:** Univariate analysis.

Variable	Cases N° (%) or Median (Range)	Controls Number (%) or Median (Range)	*p*
Gender			0.933
*Female*	248 (64.6)	253 (64.9)	
*Male*	136 (35.4)	137 (35.1)	
Marital status			0.972
*Divorced*/*Separated*/*Single*/*Widower*	223 (58.1)	226 (57.9)	
*Married*/*Living with a partner*	161 (41.9)	164 (42.1)	
Children			0.438
*Yes*	203 (52.9)	217 (55.6)	
*No*	181 (47.1)	173 (44.4)	
State of good health			0.176
*Yes*	337 (87.8)	354 (90.8)	
*No*	47 (12.2)	36 (9.2)	
Professional role			0.078
*Doctor*	159 (41.4)	133 (34.1)	
*Nurse*	140 (36.4)	143 (36.7)	
*Midwifery*	6 (1.6)	6 (1.5)	
*Ancillary worker*	52 (13.5)	79 (20.3)	
*Technician*	12 (3.1)	16 (4.1)	
Anxiety	5 (0–19)	6 (0–20)	0.055
Depression	2 (0–18)	3 (0–20)	<0.001
Decision latitude	50 (38–68)	48 (34–78)	0.108
Job demand	24 (4–43)	23 (7–43)	0.060
Personal burnout	33 (0–100)	37 (0–96)	0.072
Work burnout	29 (0–96)	32 (0–96)	0.072
Client burnout	12.5 (0–83.3)	12.5 (0–100)	0.110

**Table 3 jcm-11-04434-t003:** Bivariate analysis.

	Anxiety	Depression	Job Demand	Decision Latitude	Personal Burnout	Work Burnout	Client Burnout	Age
Anxiety	1.000	0.666 **	0.074 *	0.064	0.675 **	0.569 **	0.321 **	0.114
*p*		<0.001	0.043	0.085	<0.001	<0.001	<0.001	0.002
Depression	0.666	1.000	0.083 *	0.034	0.633 **	0.583 **	0.365 **	0.195
*p*	<0.001		0.022	0.365	<0.001	<0.001	<0.001	<0.001
Job demand	0.074	0.083 *	1.000	0.364 **	0.210 **	0.381 **	0.295 **	−0.101 *
*p*	0.043	0.022		<0.001	<0.001	<0.001	<0.001	0.006
Decision latitude	0.064	0.034	0.364 **	1.000	0.118 *	0.175 **	0.081 *	−0.324
*p*	0.085	0.365	<0.001		0.002	<0.001	0.039	<0.001
Personal burnout	0.675 **	0.633 **	0.210 **	0.118	1.000	0.755 **	0.429 **	0.147
*p*	<0.001	<0.001	<0.001	0.002		<0.001	<0.001	<0.001
Work burnout	0.569 **	0.583 **	0.381 **	0.175 **	0.755 **	1.000	0.553 **	0.131 *
*p*	<0.001	<0.001	<0.001	<0.001	<0.001		<0.001	<0.001
Client burnout	0.321 **	0.365	0.295 **	0.081 *	0.429 **	0.553 **	1.000	0.151 **
*p*	<0.001	<0.001	<0.001	0.039	<0.001	<0.001		<0.001
Age	0.114	0.195 *	−0.101 *	−0.0324 **	0.147 *	0.131 *	0.151 **	1.000
*p*	0.002	<0.001	0.006	<0.001	0.048	<0.001	<0.001	

* *p* < 0.05; ** *p* < 0.01.

**Table 4 jcm-11-04434-t004:** Multivariate analysis.

Variable	Depression	Anxiety	Personal Burnout	Work Burnout	Client Burnout	Job Demand	Decision Latitude
	Β*(p)	Β*(p)	Β*(p)	Β*(p)	Β*(p)	Β*(p)	Β*(p)
Personal burnout	0.155 (0.001)	0.417 (<0.001)		0.544 (<0.001)	−0.104 (0.048)	−0.094 (0.098)	
Work burnout	0.127 (0.006)		0.562 (<0.001)		0.547 (<0.001)	−0.094 (0.098)	0.105 (0.009)
Client burnout	0.093 (0.006)		−0.068 (0.016)	0.264 (<0.001)		0.114 (0.008)	
Job demand				0.164 (<0.001)	0.095 (0.007)		0.310 (<0.001)
Decision latitude		−0.049 (0.072)		0.045 (0.067)		0.288 (<0.001)	
Age	0.073 (0.08)	−0.070 (0.012)		0.059 (0.017)	0.144 (<0.001)		−0.268 (<0.001)
State of good health	−0.70 (0.014)	−0.060 (0.030)					−0.065 (0.084)
Doctor	−0.045 (0.099)		−0.061 (0.048)	0.095 (0.003)			
Nurse/Midwifery			−0.075 (0.016)	0.080 (0.012)		0.236 (<0.001)	−0.166 (<0.001)
Having children					−0.090 (0.024)		−0.96 (0.033)
Anxiety	0.453 (<0.001)		0.325 (<0.001)				
Depression			0.113 (0.001)	0.092 (0.002)		−0.100 (0.030)	
Cases controls	−0.053 (0.043)	0.430 (<0.001)			−0.053 (0.090)	0.083 (0.014)	
R^2^	0.589	0.607	0.706	0.720	0.416	0.319	0.252

B* = beta coefficient.

## Data Availability

Data are available upon request to the author for correspondence.

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
