# Peer review of "Assessment of Anxiety, Depression, Work-Related Stress, and Burnout in Health Care Workers (HCWs) Affected by COVID-19: Results of a Case–Control Study in Italy"

_jcm, 2022, doi:10.3390/jcm11154434_

Round 1
Reviewer 1 Report
This is a well-written article that addresses an important issue. My comments are all fairly minor, as the authors did a great job with placing their study in the context of the current literature.
1) The abstract is confusing until the reader reaches the Discussion section, since it appears in the abstract that the case group is doing better than the control group (with lower rates of depression). As a result, it is confusing, without the authors mentioning the potential reasons for the lower rates of depression, to have them stress the need for additional mental health support. I believe this could be clarified in the abstract so that the conclusion feels logical at that stage as well.
2) The Discussion claims there was "no significant difference in anxiety" between groups, however the authors presented between group differences in the univariate analyses. This statement should be updated to accurately reflect the findings.
3) The Conclusion states that there's a need for "mental health prevention among healthcare professionals" but in my opinion does not go far enough; it would be helpful to provide more concrete solutions for what steps should be taken to better support the mental health of our clinicians. Here are some examples from the literature that could be incorporated (or other ideas that the authors have for what they would like to see done would advance this important work):
https://doi.org/10.1016/j.psychres.2021.113878
https://doi.org/10.7326/M20-4199
https://doi.org/10.1016/j.jpsychores.2020.110112
Minor comments related to typos/English language/clarifying writing:
Line 23: should be "from the Occupational Medicine..."
line 24: should be "who were administered"
Line 53: should be "pandemic" [singular]
Line 70: should be "interventions" [plural]
Line 69: would be helpful as a reader if you create a new paragraph starting with "The introduction of social..."
Line 77: would be helpful as a reader if you create a new paragraph starting with "According to the job..."
Line 98: should be "lower levels" (not less); also depression should not be capitalized
Line 105: should be "Seong and colleagues"
Section 2.3: I'm used to seeing periods instead of commas in the stats
Line 274: "are not made by HCWs" sounds strange in English; could replace with "are not composed of"
Thanks for this important work!
Reviewer 2 Report
Numerous similar studies have already been published. Authors should provide more justification to help readers follow the present study's rationale and procedures.
Specific comments:
1. The abstract should be shortened to approximately 200 words as per the journal's guidelines.
2. Please change "Civil status" to "Marital status".
3. What was the estimated response rate of the present study? A low response rate can contribute to non-response bias or a negative bias among respondents.
4. The job roles in Table 1 are rather confusing. Why are "Nurse / Obstetrician" classified in the same category? Do you mean midwife or an obstetrician, which is usually a doctor by training? "Other health care worker" is too broad and unhelpful for the purposes of the present study and analysis. There is about 4% of missing data, which is to be expected. "Missing" data should be excluded.
5. In the univariate analysis in Table 2, somehow "Doctor / Nurse / Obstetrician" are now lumped under the same category, which is different from the categories used in the multivariate analysis in Table 4. This is very inconsistent and confusing for the reader.
6. For intergroup comparisons, please state the level of significance.
7. "The tests were administered in the presence of a doctor, a condition which may have influenced the HCWs response" - why was this so? Was this really necessary for data collection? At least some justification and explanation are needed. Was the doctor also a senior staff member of the hospital? Social desirability bias would play a big factor here.
8. Some comments on the local COVID-19 situation on the ground during the period the survey was conducted would be helpful. If the study was implemented during peak COVID periods, the findings may have been an anomaly (but may not be of course). It may be useful to plot the trend in cases over time as well so readers have a better understanding of the context.
9. In the discussion section, as part of mitigation strategies, as resources could be particularly scarce during a serious pandemic situation, timely psychological support could also take many forms, including telemedicine and informal support groups (citation: pubmed.ncbi.nlm.nih.gov/32380875). This should be mentioned.
10. The conclusion should be condensed as a single cogent paragraph.
Reviewer 3 Report
The reviewer was pleased to read the manuscript submitted for review under the title “ Assessment of Anxiety, Depression, Work-Related Stress and 2 Burnout in Health Care Workers (HCWs) affected by COVID- 3 19: Results of a Case-control Study in Italy “
The quality of presentation of the manuscript’s topic, the interest of the readers and the scientific soundness are all highly appreciated.
It is recommended to improve the text and / or explain the answers to the problems reported by the reviewer in detail:
-
line 15-16: Correspondence person's data should include email address only. Please remove full postal address and first and last name.
-
the questionnaire included three tests: Hospital Anxiety and Depression Scale 138 (HADS), Copenhagen Burnout Inventory (CBI) and Karasek’s Job Content Questionnaire 139 (JQC) should be attached as supplemental materials to the main text.
-
line 318-321 ,missing period between sentences, (quote) “From an occupational point of view and the results are 318 indicating the need for a workplace health promotion activities based on stress and burn- 319 out management and prevention There is sufficient evidence that multiple organizational 320 and work-related interventions can lower the impact of mental-health related issues in the 321 COVID-19 pandemics. “
Round 2
Reviewer 2 Report
Thank you for the revisions.
Could the authors clarify what is meant by "the estimated response rate was not an issue, since the filling of the questionnaire was mandatory and included in the medical surveillance of HCWs both for cases and controls"? According to your manuscript, there are some 4500 HCWs working in the hospital but only a total of 774 participants were enrolled in your study despite the filling of the questionnaire being made mandatory? It is also important to state factually that this was a surveillance study for all staff in the hospital if this was indeed the case.
Author Response
Many thanks for this comment, so we can clarify.
This is a case-control study in which all cases of SARS-Cov-2 among hospital staff were recruited in the study period, and controls were recruited after matching for age and gender.
The following statement was added in the Results section.
"The response rate was 100% for both groups, since the questionnaires were administered during the surveillance visit for all staff in the hospital and filling them was mandatory."